# Tympanoplasty and adenoidectomy in children: Comparison of simultaneous and sequential approaches

**Timur Shamshudinov[1], Laura Kassym[2]\*, Saule Taukeleva[3], Bolat Sadykov[1], Hassan Diab[4], Mario Milkov[5]**

**1** Center of Pediatric Otorhinolaryngology, General Hospital #5, Almaty, Republic of Kazakhstan, **2** School of Medicine, Nazarbayev University, Nur-Sultan, Republic of Kazakhstan, **3** Kazakh-Russian Medical University, Almaty, Republic of Kazakhstan, **4** The National Medical Research Center for Otorhinolaringology, Federal Medico-Biological Agency, Moscow, Russian Federation, **5** Medical University of Varna, Faculty of Dental Medicine, Varna, Bulgaria

\* laura.kassym@gmail.com

**Data Availability Statement:** All relevant data are within the paper and its Supporting Information files.

## Abstract

### Background

The authors sought to compare simultaneous and sequential tympanoplasty and adenoidectomy surgery in pediatric patients.

### Methods

This retrospective single-center study included 65 children (36 males, 29 females; mean age 9.16 ± 3.82 years; range 3–17 years) requiring both tympanoplasty and adenoidectomy. Simultaneous surgeries were performed on the same day, during single general anesthesia, whereas sequential surgeries were separated at least 12 weeks. The groups were compared with regard to restoration of hearing, tympanic membrane status, and utilization of medical resources. All study participants had a 12-months follow-up period after surgery.

### Results

No statistically significant differences were observed between the groups regarding pre- and post-operative ABG values and average hearing gains. However, the post-operative ABG was significantly lower than the pre-operative ABG in both groups (p<0.001). There were no significant differences between simultaneous and sequential groups with respect to complete healing rates and complications (all p>0.355). Simultaneous tympanoplasty and adenoidectomy surgery management is associated with a significantly decreased cumulative hospital stay, cumulative operating room time, and cumulative pure surgical time (all p≤0.016).

### Conclusions

The results of first comparative study of simultaneous versus sequential tympanoplasty and adenoidectomy surgery managements demonstrate no advantages for the sequential

**Funding:** The author(s) received no specific funding for this work.

**Competing interests:** The authors have declared that no competing interests exist.

approach. The same-day surgery can show the clinical outcomes comparable to those in the sequential group. The simultaneous surgery approach appears to be associated with reduced medical resources consumption. Therefore, simultaneous surgery management is an effective and safe option for children with chronic otitis media and adenoid hypertrophy.

## Background

Chronic otitis media (COM) is one of the most common otological conditions in childhood [1]. COM is the essential factor of hearing loss, resulting in impairment of speech development, behavior, academic performance, and social competence [2]. The evidence from literature demonstrates that COM affects negatively children's and caregivers' well-being and quality of life [3].

Adenoid hypertrophy is thought to play an overriding role in the pathogenesis of otitis media. Firstly, adenoids may serve as a reservoir for bacterial and viral infection. Secondly, adenoids lie closely to the orifices of the Eustachian tube leading to disruption of ventilation, protection, and clearance of middle ear. Notably, Eustachian tube disfunction detected with semi-objective (Eustachian tube score; ETS-7) and objective (computed tomography combined with Valsalva maneuver) methods is associated with high prevalence and incidence of COM in children [4,5]. Microorganisms persisting in adenoids may spread through the dysfunctional Eustachian tube and cause acute or chronic inflammation. Also, both adenitis and otitis media are already recognized as biofilm infections [6].

From that perspective, adenoidectomy seems to be an optimal solution in the management of otitis media. Currently, adjuvant adenoidectomy is recommended as a beneficial intervention in children who are older than 4 years old and who had one or repeated tympanostomy tube insertions [7]. There are several studies available in the literature assessing the contribution of adenoidectomy in the successful treatment of children with OM. The first prospective randomized controlled study presented by Maw in 1984 demonstrated that adenoidectomy has a significant therapeutic effect in resolving established otitis media with effusion in 36–48% of cases for up to 12 months [8]. Recent systematic reviews demonstrated that adjuvant adenoidectomy has a protective role for recurrent acute otitis media, persistent otitis media with effusion, and otorrhea [7,9]. However, the vast majority of articles have reported the results of adjuvant adenoidectomy in children with otitis media with effusion. The adjuvant adenoidectomy permits to decrease the number of repeat tympanostomy tube insertions in comparison with tympanoplasty alone [10,11]. The recent study of Ferlito et al. demonstrated that concurrent adenoidectomy was the contributing factor to the lower incidence of postoperative otorrhea in children with OME [12]. Also, the relation between adenoids and Eustachian tube in etiopathology of OM has been widely discussed. Among the few clinical studies, some reported the significance of adenoids size, while others noted the relevance of adenoids abutment to torus tubarius [13–15]. By some authors, adjuvant adenoidectomy seems to be the cost-efficient intervention which allows to decrease the number of rehospitalizations [6,16]. Moreover, even the enhancement of Eustachian tube function alone may contribute significantly to the curative process of OME. Balloon eustachian tympanoplasty combined with myringotomy and tube insertion was recognized as efficient intervention in the treatment of OME in children [17].

Therefore, most prior publications addressed to issues of the initial adjuvant adenoidectomy in the management of OM. There are no previous studies or reports of simultaneous

adenoidectomy and tympanostomy surgery in the pediatric population. This article compares surgical outcomes, hearing improvement rates, and utilization of medical resources between patients for whom adenoidectomy and tympanostomy were performed either simultaneously or sequentially.

## Methods

This retrospective, comparative, non-randomized clinical study was conducted between 03/2018 and 03/2020 at the Ear Nose Throat department of General Hospital #5 of Almaty City, Kazakhstan. The study was approved by the Ethics Committee of Kazakh medical university of continuing education (Protocol no.: 1, from January 15, 2019), and the research was conducted in compliance with principles of the Declaration of Helsinki and the Guideline for Good Clinical Practice. The written informed consent was obtained consent from parents or guardians of participant. The participants who were 15 or elder at the moment of enrollment gave the written informed consent themselves.

The study population consisted of 65 patients (36 males, 29 females; mean age 9.16±3.82 years; range 3–17 years) who underwent power-assisted (microdebrider) adenoidectomy and type 1 tympanoplasty. Children were included if they had been followed up at our clinic for at least 12 months. All forms of chronic otitis media and all degrees of adenoid hypertrophy were included. The exclusion criteria were as follows: subjects who were younger than 3 years old and older than 18 years old; who had contraindications for general anesthesia; who had the following disorders including active suppurative inflammatory process ear, nose, or throat, and cholesteatoma. Also, patients who did not at the regular follow-up visits were excluded (see S1 Fig).

All patients underwent preoperative temporal computed tomography (CT). Temporal CT evaluated middle ear pathologies and neurovascular anomalies. The middle ear pathologies were observed endoscopically from tympanic membrane perforation prior to surgery. The video-otoscopy was performed for the detection of the size of membrane perforation before and after the operation. Pure tone audiometry was conducted for the evaluation of pre- and post-operative hearing function. The diagnosis of adenoid hypertrophy was established by patient's history and nasopharyngeal endoscopy. The degree of adenoid hypertrophy was assessed according to Clemens and McMurray scale, i.e. grade 1 ($< 1/3^{rd}$ choanal obstruction), grade 2 ($1/3^{rd}$ to $2/3^{rd}$ choanal obstruction), grade 3 ($2/3^{rd}$ to almost complete choanal obstruction by the adenoid tissue) and grade 4 (complete obstruction) [18]. All the children were operated under general anesthesia, conducted according to the well-established principles of ENT surgery.

Age, gender, status and pathologies of the adenoids and the middle ear, size of perforation in the tympanic membrane, preoperative and postoperative audiological test results, the condition of tympanic membrane and complications as observed during the postoperative follow-up periods, consumption of medical sources including time of hospitalization and surgery were recorded. All data were collected by reviewing the patients' files in the hospital database. All patients provided informed written consent to have data from their medical records for use in research.

The children included in the study were assigned to two groups according to the type of surgery organization. All patients prior to 01/2019 underwent sequential (separate day) adenoidectomy and tympanoplasty surgery, whereas all patients after 01/2019 were managed with simultaneous (same day) adenoidectomy and tympanoplasty surgery. In our hospital, all of the conventional adenoidectomies were performed using general anesthesia with endotracheal intubation. The patients were placed in the Rose position by putting a pillow under their

shoulders. A McIvor surgical retractor was used as a mouth gag. The nasal cavity and naso-pharynx were evaluated using a 45˚ angled 11-cm rigid fiber-optic endoscope with a video attachment (HOPKINS, Karl Storz, Germany). A red rubber catheter was inserted through the nasal cavity to retract the soft palate and to visualize better all parts of oropharynx. Microdebri-der procedure was performed using 12-cm UNIDRIVE microdebrider console with urved blade as well as 65-degree angulation blade (Karl Storz, Germany). Adenoid tissue was excised starting from the vomer towards to palatoglossal fold through the choanal lateral areas. Then increased tubal tonsils were excised from fossae of Rosenmuller using a 45˚ angled curved blade in top-to-bottom direction medially. Fossae of Rosenmuller were emptied totally for free movements of torus tubarius.

All the tympanoplasties were performed with the patient under general anesthesia with endotracheal intubation. The grafts were harvested from the temporal fascia muscle by a 2-3-cm supraaural incision. Graft tissue was placed on the freshened edges of perforation of tym-panic membrane for reconstruction with an onlay technique. The assessment of tympanic membrane and ossicular chain was performed using a rigid endoscope with an outer diameter of 3 mm and a length of 11 cm (Hopkins, Karl Storz, Germany) at an angle of 30 degrees. The images were viewed on 27-inch Full HD monitor. All infected, destructive, and other patholog-ical tissues in the tympanic cavity were removed using endoscope and curved microneedle. If the defect of ossicular chain was detected during surgery a concomitant ossiculoplasty was performed.

All patients were prescribed a course of oral antibiotics during postoperative 1 week (type of antibiotic was selected according to patients' sensitivity). Additionally, non-steroidal anti-inflammatory drugs (acetaminophen, ibuprofen, ketoprofen) were given in all children in case of fever or pain.

The postoperative air-bone gap (ABG) values and the integrity of the tympanic membrane were assessed at 1, 3, 6, and 12 months of follow-up period. The groups were compared according to the pre- and post-operative ABG values and to the post-operative status of tym-panic membrane (percentage of patients with unhealed tympanic membrane). Also, we ana-lyzed medical services utilization of the two groups, including average admission and surgery time and drug consumption.

## Statistical analysis

The R 3.6.3 software (R Foundation for Statistical Computing, Vienne, Austria) was used for statistical analyses. The data were analyzed using descriptive statistics (mean and standard deviation, frequency). The comparison of qualitative data between two groups was performed using a Mann-Whitney U test. Categorical variables were analyzed using Fisher's exact test. Postoperative and preoperative evaluation results were compared using Friedman test. For post-hoc analysis the Conover test was used. A p-value <0.05 was defined as statistically significant.

## Results

Thirty patients (46%) had simultaneous surgery during a single anesthesia session. Thirty-five children (54%) had sequential surgery during two anesthesia sessions, separated by average 25.48 ± 9.44 weeks. In all cases, adenoidectomy was performed prior to tympanoplasty. The ossiculoplasty was performed in three and four patients of simultaneous and sequential groups respectively. The two groups did not significantly differ in age, gender, grade of adenoid hyper-trophy, localization, and size of perforation (all *p* >0.05; Table 1).

**Table 1. Comparison of subject data in simultaneous and sequential groups.**

| Variables | Simultaneous (n = 30) | Sequential (n = 35) | *p* |
|---|---|---|---|
| **Age (years)** | 8.73±3.80 | 9.54±3.86 | 0.469* |
| **Gender** | | | |
| **Females** | 15 (50%) | 14 (40%) | 0.461** |
| **Males** | 15 (50%) | 21 (60%) | |
| **Adenoid hypertrophy** | | | |
| **Grade 1** | 5 (16.7%) | 5 (14.3%) | 0.948** |
| **Grade 2** | 9 (30.0%) | 12 (34.3%) | |
| **Grade 3** | 5 (16.7%) | 5 (14.3%) | |
| **Grade 4** | 11(36.6%) | 13 (37.1%) | |
| **The side operated** | | | |
| **Left ear** | 13 (43.3%) | 20 (57.1%) | 0.901** |
| **Right ear** | 17 (56.7%) | 15 (42.9%) | |
| **Perforation size** | | | |
| **Total** | 18 (60%) | 22 (62,9%) | 0.813** |
| **Subtotal** | 12 (40%) | 12 (37,1%) | |

Note.

*—Mann-Whitney U test

**—Fisher's exact test.

The mean pre-operative and post-operative (after 12 months) ABG in the simultaneous group were 44.80±9.68 and 9.37±9.96 dB, respectively; the corresponding values in the sequential group were 44.91±9.42 and 7.73±11.45 dB. The ABG values showed a significant reduction in both groups (p<0.0001). The post-operative ABG values at 1, 3 and 6 months of follow-up period did not differ statistically significant between two groups (all p>0.05). Also, we did not detect any statistically significant differences between the groups with the respect to the average hearing gain (p = 0.481; Table 2).

We also assessed the rates of successful anatomical restoration of the tympanic membrane in two groups. The complete healing rate was 93.3% in the simultaneous group and 94.3% in the sequential group; the difference was not significant (p≈1). No complications such as mastoiditis, loss of blood, persistent otorrhea, sore throat, fever, or dizziness were detected during

**Table 2. Improvement of hearing function.**

| | Group | | *p* |
|---|---|---|---|
| | Simultaneous (n = 30) | Sequential (n = 35) | |
| **Pre-operative ABG** | 44.80 (9.68) | 44.91 (9.42) | 0.884* |
| **Post-operative ABG (after 1 month)** | 31.85 (7.63) | 32,01 (10.68) | 0.747* |
| **Post-operative ABG (after 3 months)** | 22.74 (9.22) | 20.87 (11.07) | 0.514* |
| **Post-operative ABG (after 6 months)** | 14.13 (11.01) | 11.93 (12.55) | 0.253* |
| **Post-operative ABG (after 12 months)** | 9.37 (9.96) | 7.73 (11.45) | 0.238* |
| *p* | <0.0001** | <0.0001** | |
| **Hearing gain (within 12 months)** | 37.18±12.15 | 35.43±12.6 | 0.481* |

Note.

*- Mann-Whitney U test: Comparison of simultaneous and sequential groups

**- Conover test: Comparison ABG between pre- and post-operative periods.

**Table 3. Surgical outcome and complication rates.**

| | Group | | p |
|---|---|---|---|
| | Simultaneous (n = 30) | Sequential (n = 35) | |
| **Complete tympanic membrane healing** | 28 (93.3%) | 33 (94.3%) | ≈1* |
| **Moderate pain** | 0 (0%) | 1 (2.86%) | 0.355* |
| **Severe pain** | 0 (0%) | 1 (2.86%) | |

Note. $p^*$- Fisher's exact test.

or after surgery. In the sequential group, one patient had moderate pain after adenoidectomy, and another presented severe pain syndrome after adenoidectomy too (Table 3).

Consumption of medical resources of the two groups was compared in some indicators (Table 4). In the sequential group, all the time parameters are defined as the numerical sums of the corresponding figures for each operation. The mean hospital admission in the simultaneous surgery group (4.43 ± 0.5 days) was significantly lower than that of the sequentially operated patients (8.51 ± 0.7 days). Comparing the length of the hospital stay after surgery, a significantly longer time was identified in the sequential group (6.51 ± 0.7 versus 3.43 ± 0.5, p<0.001). Total time in the operating room includes the arrival of patient in the operative room, process of anesthesia, operation's start and end, the leave of patient from operating room. Comparison of that parameter identified the simultaneous approach as significantly quicker (p<0.001) than the sequential surgical procedures. Pure surgical time was significantly lower in the simultaneously operated patients (p = 0.016).

## Discussion

In this article, the functional and anatomical success rates, and medical services consumption of sequential and simultaneous surgery managements of tympanoplasty and adenoidectomy in pediatric patients, were compared. To our knowledge, this is the first study on this subject in the literature. There is plenty of studies showing the advantages of adjuvant adenoidectomy in children with otitis media with effusion. Medical, educational, and economic benefits of adenoids surgery performed prior to tympanal tube insertion have been demonstrated already. None of the studies we analyzed were related to the comparison of combined versus sequential surgery tympanoplasty and adenoidectomy surgery.

The existing data demonstrated the curative effect of adenotonsillectomy in pediatric patients with obstructive sleep apnea syndrome [19]. Furthermore, the performance of adenotonsillectomy improved the sleep-related quality of life and behavioral parameters in children

**Table 4. Utilization of medical services.**

| | Group | | p |
|---|---|---|---|
| | Simultaneous (n = 30) | Sequential (n = 35) | |
| **Total length of hospital stay (days)** | 4.43 ± 0.5 | 8.51 ± 0.7** | <0.0001* |
| **Length of hospital stay after surgery (days)** | 3.43 ± 0.5 | 6.51 ± 0.7** | <0.0001* |
| **Total time in the operating room (min)** | 157.4 ± 20.4 | 209.1 ± 21.6** | <0.0001* |
| **Pure surgical time (min)** | 93.1 ± 10.8 | 99.9 ± 10.5** | 0.016* |

Note.

*- Mann-Whitney U test: Comparison of simultaneous and sequential groups

**—cumulative figures are given for the sequential surgery group.

with OSAS [20]. The majority of studies were conducted also to assess the results of adjuvant adenoidectomy in pediatric patients with OME. For instance, Maw compared the OME resolution rates in children after adenoidectomy and in the no-surgery group. He found the increased benefit in patients who underwent operation expressed in a number of resolved cases of established OME for up to 12 months [8]. Coyte et al. noted that the risks of readmissions and tympanal tube reinsertion were reduced significantly after the performance of adjuvant adenoidectomy [16]. Other researchers also recognized the protective role of adenoids surgery in children [6,10]. Following systematic reviews and meta-analyses reported improved clinical outcomes and the optimal age group for adenoidectomy performed prior tympanoplasty. These findings enabled the experts to publish the guidelines according to which adenoidectomy should be performed in children older than 4 years old and who had already undergone tympanostomy [21]. Nevertheless, in some unique patient groups several risk factors may contribute greatly to more repeat tube insertion (TTIs). For instance, the younger age and COME were recognized as the adjusted risk factors for TTIs in children with Down syndrome [22].

The main factor of the success of adenoidectomy prior to the tympanoplasty is the restoration of the Eustachian tube's functions. Currently it has been shown with semi-objective method (ETS-7) that children with adenoid hypertrophy have a high incidence of Eustachian tube dysfunction [23]. In normal conditions, the ET provides the ventilation, drainage, and clearance of the tympanal cavity. On the other hand, the ET dysfunction might be attributable to the range of Waldeyer 's ring disorders. Adenoids, in general, may impact on the development of OM in three ways. One of the most obvious factors in ethiology of OM is the anatomic obstruction of ET. This idea is supported with the findings of some researchers. For instance, Nguyen in 2003 described the benefits of adenoidectomy and pressure equalization tube insertion in patients with OM in case if the adenoids were adjacent to torus tubarius [24]. Skoloudik et al. also noted that the relation between adenoids and torus tubarius is much more important than the volume of lymphoid tissues [15]. Moreover, Abdel Tawab&Tabook in their recent study reported the highly significant correlation between adenoid size and mucoid nature of middle ear fluid [25]. However, the role of adenoids in the pathogenesis of OM is more sophisticated due to infectious and inflammatory aspects. The surface of Waldeyer's ring structures is colonized by both non-virulent and pathogenic microbes which may contribute greatly in the development of OM. Adenoids may act as a reservoir for chronic infection agents causing biofilm formation. In turn, bacterial biofilm is strongly associated with persistent inflammation in the middle ear and refractory forms of OM. Tawfik et al. evidenced in a group of adenoid hypertrophy and OME patients that 100% of their adenoid samples were covered by biofilms [26]. Saylam et al. found the biofilms in samples of patients who had undergone adenoidectomy only and adenoidectomy combined with ventilation tube insertion for COM. The COM group had higher-grade biofilm formation in comparison to another group [27]. The study of Bhat et al. showed that 36 of 100 pediatric with adenoid hypertrophy had the asymptomatic OME [28]. The presence of biofilms on the adenoid surface may maintain chronic illness leading to possible missed complication cases. These study results, therefore, support the utilization of adjunct adenoidectomy as a beneficial and effective intervention in the current management of pediatric patients with COM.

During this study, 65 tympanoplasties combined with adenoidectomy were performed over a period of two years. The first group of study participants (n = 30) was subjected to simultaneous surgery, whereas the second group (n = 35) was subjected to sequential surgical protocol. In all cases in this study adenoidectomy was performed prior to tympanoplasty. There were no statistical differences between the groups regarding patient's age, gender, and other baseline characteristics. We observed no statistical differences in the pre-operative (44.8 versus

44.91 dB, p>0.05) and post-operative ABG (9.37 versus 7.73 dB, p>0.05) regardless of which surgical protocol was utilized. The rates of hearing restoration within follow-up periods and average hearing gain were similar between the groups. In both surgery managements, the significant hearing improvement may be attributed to the adjuvant role of adenoidectomy. The additional benefit from adenoidectomy expressed as better hearing in children with OME was described in other studies. In 1993, Maw&Bawden reported that hearing function was improved better in case of larger adenoid size (13.1±2.6 dB per cm size of adenoids removed) [29]. MRC Multicentre Otitis Media Study Group described that adenoidectomy extended the better hearing through the second year in children with OME at least 20 dB hearing level in both ears [30]. Therefore, we may conclude that simultaneous surgery demonstrates similar results to sequential management for the improvement of hearing function.

Successful tympanic membrane healing is another essential parameter by which the efficiency of both approaches can be assessed. Same-day surgery was equally efficacious regarding successful anatomical restoration over 12 months follow-up period (93.3% versus 94.3%, p≈1). Salviz et al. reported that adenoid disease adversely affected the successful anatomical outcome rate in patients who underwent the type I tympanoplasty using temporalis fascia graft. Only twelve patients achieved intact tympanic membrane without any retraction or lateralization for at least 6 months after operation [31]. Also, our analysis has shown that post-operative complications were rare. Thus, we may conclude that simultaneous surgical procedure is on average no riskier than the sequential approach. Finally, the present study demonstrates that utilization of simultaneous management allows to reduce both hospital stay and operative time. The simultaneous adenoidectomy and tympanoplasty require only one hospital admission, one anesthesia induction, and shorter surgery time. Some experimental and clinical studies suggested that repeated general anesthesia may lead to subsequent learning disabilities and behavioral abnormalities in young animals and young children [32,33]. Therefore, the simultaneous management seems economically more attractive by shortening primary (hospital stays, surgery times, and anesthesia episodes) and secondary (sick leaves) healthcare costs.

To date, our study is the first comparing simultaneous and sequential surgery organizations of adenoidectomy and tympanoplasty in the pediatric population. Our data demonstrate that same-day surgery can provide sufficient outcomes that are at least as safe as those following sequential operations. Nonetheless, there are some drawbacks to this study. Firstly, it is a retrospective nature of comparison with a relatively small number of participants. This shortage was compensated by longer follow-up assessments which are sufficient for full-value evaluation of both anatomical and functional conditions. However, a multi-center study with a larger study population would be beneficial. Another limitation of our study is the absence of randomization which may serve as a resource of selection bias. The impact of non-randomized approach was mitigated by consequent recruitment of patients with a precise temporal division (before and since January, 2019). Also, there were no statistically significant differences regarding the most important baseline characteristics of study participants. We, therefore, believe that any discrepancies between the groups were avoided because all surgical procedures were performed by the senior author alone.

## Conclusions

The present study shows that the children who had undergone the same day adenoidectomy and tympanoplasty achieve anatomical and functional outcomes as fast as those who were exposed to the sequential approach. Despite the alleged increased risks from simultaneous surgery, no complications were identified in our study. The simultaneous surgical protocol appears to be a clinically efficient and safe operative procedure with some advantages. A

shorter hospital stay and a single anesthetic course might seem very beneficial for pediatric patients and caregivers. However, further prospective controlled studies conducted with larger study populations are needed to confirm our conclusions.

## Supporting information

**S1 Fig. Flow chart of study design.**
(TIF)

**S1 File.**
(XLSX)

## Author Contributions

**Conceptualization:** Saule Taukeleva, Bolat Sadykov, Hassan Diab, Mario Milkov.

**Data curation:** Timur Shamshudinov, Saule Taukeleva, Bolat Sadykov.

**Formal analysis:** Laura Kassym.

**Methodology:** Timur Shamshudinov, Laura Kassym.

**Project administration:** Timur Shamshudinov, Laura Kassym, Saule Taukeleva, Bolat Sadykov.

**Resources:** Timur Shamshudinov, Saule Taukeleva, Bolat Sadykov, Hassan Diab, Mario Milkov.

**Software:** Timur Shamshudinov, Laura Kassym, Hassan Diab.

**Supervision:** Laura Kassym, Saule Taukeleva, Bolat Sadykov, Hassan Diab, Mario Milkov.

**Validation:** Timur Shamshudinov, Laura Kassym.

**Visualization:** Laura Kassym.

**Writing – original draft:** Timur Shamshudinov, Laura Kassym, Mario Milkov.

**Writing – review & editing:** Timur Shamshudinov, Laura Kassym, Saule Taukeleva, Bolat Sadykov, Hassan Diab, Mario Milkov.

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
