## [Decision Letter · Decision Letter 0]

25 Nov 2021

PONE-D-21-32269Tympanoplasty and Adenoidectomy in Children: Comparison of Simultaneous and Sequential ApproachesPLOS ONE

Dear Dr. Kassym,

Thank you for submitting your manuscript to PLOS ONE. After careful consideration, we feel that it has merit but does not fully meet PLOS ONE’s publication criteria as it currently stands. Therefore, we invite you to submit a revised version of the manuscript by Jan 09 2022 11:59PM that addresses the points raised during the review process .

We look forward to receiving your revised manuscript.

Kind regards,

Academic Editor

PLOS ONE

Reviewers' comments:

Reviewer's Responses to Questions

**Comments to the Author**

1. Is the manuscript technically sound, and do the data support the conclusions?

Reviewer #1: Yes

Reviewer #2: Yes

2. Has the statistical analysis been performed appropriately and rigorously? 

Reviewer #1: Yes

Reviewer #2: Yes

3. Have the authors made all data underlying the findings in their manuscript fully available?

Reviewer #1: Yes

Reviewer #2: Yes

4. Is the manuscript presented in an intelligible fashion and written in standard English?

Reviewer #1: No

Reviewer #2: Yes

5. Review Comments to the Author

Reviewer #1: The topic of the paper is interesting and sound scientifically relevant. However, some major corrections have to be performed to improve its quality before publication.

Background:

Line 5: In children, Eustachian Tube is less functional than the adult one and it depends on its different anatomical position. Eustachian tube function has been evaluated with semi-objective method (ETS-7) and objective one (CT) that gives information about its anatomical-functional activity and the relation with COM (atelectasis, OME, and cholesteatoma). cite https://doi.org/10.1016/j.anorl.2017.11.009; doi: 10.1371/journal.pone.0247708.

Methods:

Line 2: You said that the retrospective study was conducted from 03/2018 to 03/2021, but it is in contrast with sentences reported in line 61 (follow up at 12 months) and line 6 of “results” where you define preoperative and postoperative ABG after 12 months. Have you made a mistake? Was it 2020? please clarify this point or define patients that have ended the follow-up and patients that are at about 6 months.

Line 41: How were patients medically treated after surgery? Please add pharmacological protocol performed (es: doi: 10.14639/0392-100X-N0150).

Results:

Line 6: add how many patients performed contemporary tympanoplasty and ossiculoplasty.

Discussion:

Line 32: Currently it has been shown with semi-objective method (Eustachian Tube Score 7) that children with Adenoid Hypertrophy, have a high incidence of Eustachian Tube Dysfunction (cite doi:10.1177/0145561321989455).

Table 3: Line of complication add “0” or “/” in the empty spaces.

References: Correct the references (1-3 are reported twice)

Reviewer #2: Minor suggestions:

Background

Line 3, change studies with evidence from literature demonstrates...

line 19, Although a greater number of sequelae is described in the use of the ventilation tube,the correct stratification of the patients allows the most appropriate approach depending on both the clinical and instrumental presentation. The surgical method is affected by the operator's experi-

ence. Other factors that infuence prognosis are the age of the patient at the time of initial placement of the tube, concomitant adenoidectomy, the type of ventilation tube, the duration of maintenance in the ventilation tube, regular follow-up by the same otologist and strict water precautions. please cite Ferlito S, Cocuzza S, Grillo C. Complications and sequelae following tympanostomy tube placement in children with effusion otitis media: Single center experience and review of literature. Acta Medica Mediterranea 36(3). doi:10.19193/0393-6384_2020_3_298.

line 29, Myringotomy and tube insertion combined with balloon eustachian tuboplasty has been demonstrated an effective and safe in the treatment of children with OME. Directly benefit from the ventilation tube, the curative effect was close during the period of tube retention. please discuss and cite Chen S, Zhao M, Zheng W, Wei R, Zhang B, Tong B, Qiu J. Myringotomy and tube insertion combined with balloon eustachian tuboplasty for the treatment of otitis media with effusion in children. Eur Arch Otorhinolaryngol. 2020 May;277(5):1281-1287. doi: 10.1007/s00405-020-05828-9. Epub 2020 Jan 30. PMID: 32002612.

Methods

- apply the consort guidelines

- perform a flow diagram describing the protocol

- describe clearly selection criteria

Results

- clear and well written

- always report both percentage and number or value

Discussion

- Adenotonsillectomy have been reported by several authors as a curative approach in children for both sleep apnea and behavioural disorders. Please discuss and cite doi: 10.1016/j.ijporl.2005.08.008. and doi:10.3390/children8100921.

- Children with down syndrome demonstrated to undergo more repeat tube insertion (tti )if they were of younger age and if the indication for surgery up to 61%. please cite doi:10.1016/j.ijporl.2021.110811.

6. PLOS authors have the option to publish the peer review history of their article (what does this mean?). If published, this will include your full peer review and any attached files.

Reviewer #1: No

Reviewer #2: No

---

## [Author Response · Author response to Decision Letter 0]

9 Jan 2022

PONE-D-21-32269

Tympanoplasty and Adenoidectomy in Children: Comparison of Simultaneous and Sequential Approaches

PLOS ONE

Dear Dr. Kassym,

Thank you for submitting your manuscript to PLOS ONE. After careful consideration, we feel that it has merit but does not fully meet PLOS ONE’s publication criteria as it currently stands. Therefore, we invite you to submit a revised version of the manuscript by Jan 09 2022 11:59PM that addresses the points raised during the review process .

We look forward to receiving your revised manuscript.

Kind regards,

Academic Editor

PLOS ONE

All files revised

All the mentioned details on ethics were added to the manuscript text.

The database was added as Supporting Information file (excel table)

All references were checked

Reviewers' comments:

Reviewer's Responses to Questions

Comments to the Author

1. Is the manuscript technically sound, and do the data support the conclusions?

Reviewer #1: Yes

Reviewer #2: Yes

2. Has the statistical analysis been performed appropriately and rigorously?

Reviewer #1: Yes

Reviewer #2: Yes

3. Have the authors made all data underlying the findings in their manuscript fully available?

Reviewer #1: Yes

Reviewer #2: Yes

4. Is the manuscript presented in an intelligible fashion and written in standard English?

Reviewer #1: No

Reviewer #2: Yes

5. Review Comments to the Author

Reviewer #1: The topic of the paper is interesting and sound scientifically relevant. However, some major corrections have to be performed to improve its quality before publication.

Background:

Line 5: In children, Eustachian Tube is less functional than the adult one and it depends on its different anatomical position. Eustachian tube function has been evaluated with semi-objective method (ETS-7) and objective one (CT) that gives information about its anatomical-functional activity and the relation with COM (atelectasis, OME, and cholesteatoma). cite https://doi.org/10.1016/j.anorl.2017.11.009; doi: 10.1371/journal.pone.0247708.

Done. We added the statement using these references.

Notably, Eustachian tube disfunction detected with semi-objective (Eustachian tube score; ETS-7) and objective (computed tomography combined with Valsalva maneuver) methods is associated with high prevalence and incidence of COM in children [4,5].

Methods:

Line 2: You said that the retrospective study was conducted from 03/2018 to 03/2021, but it is in contrast with sentences reported in line 61 (follow up at 12 months) and line 6 of “results” where you define preoperative and postoperative ABG after 12 months. Have you made a mistake? Was it 2020? please clarify this point or define patients that have ended the follow-up and patients that are at about 6 months.

Thank you. We corrected this technical mistake

This retrospective, comparative, non-randomized clinical study was conducted between 03/2018 and 03/2021 03/2020 at the Ear Nose Throat department of General Hospital #5 of Almaty City, Kazakhstan.

Line 41: How were patients medically treated after surgery? Please add pharmacological protocol performed (es: doi: 10.14639/0392-100X-N0150).

Thank you. We added the pharmacological treatment to the text of manuscript.

All patients were prescribed a course of oral antibiotics during postoperative 1 week (type of antibiotic was selected according to patients’ sensitivity). Additionally, non-steroidal anti-inflammatory drugs (acetaminophen, ibuprofen, ketoprofen) were given in all children in case of fever or pain.

Results:

Line 6: add how many patients performed contemporary tympanoplasty and ossiculoplasty.

Thank you. Done.

The ossiculoplasty was performed in three and four patients of simultaneous and sequential groups respectively.

Discussion:

Line 32: Currently it has been shown with semi-objective method (Eustachian Tube Score 7) that children with Adenoid Hypertrophy, have a high incidence of Eustachian Tube Dysfunction (cite doi:10.1177/0145561321989455).

Thank you. We added this statement to the text of manuscript.

Currently it has been shown with semi-objective method (ETS-7) that children with adenoid hypertrophy have a high incidence of Eustachian tube dysfunction [23]

Table 3: Line of complication add “0” or “/” in the empty spaces.

Thank you. We removed the subheading “Complication” related to the pain rate from table.

References: Correct the references (1-3 are reported twice)

Done

Reviewer #2: Minor suggestions:

Background

Line 3, change studies with evidence from literature demonstrates...

Thank you. We paraphrased this sentence.

The evidence from literature demonstrates that COM negatively affects children's and caregivers' well-being and quality of life

line 19, Although a greater number of sequelae is described in the use of the ventilation tube,the correct stratification of the patients allows the most appropriate approach depending on both the clinical and instrumental presentation. The surgical method is affected by the operator's experi-

ence. Other factors that infuence prognosis are the age of the patient at the time of initial placement of the tube, concomitant adenoidectomy, the type of ventilation tube, the duration of maintenance in the ventilation tube, regular follow-up by the same otologist and strict water precautions. please cite Ferlito S, Cocuzza S, Grillo C. Complications and sequelae following tympanostomy tube placement in children with effusion otitis media: Single center experience and review of literature. Acta Medica Mediterranea 36(3). doi:10.19193/0393-6384_2020_3_298.

Thank you. We added this sentence to the text of manuscript.

The recent study of Ferlito et al. demonstrated that concurrent adenoidectomy was the contributing factor to the lower incidence of post-operative otorrhea in children with OME [12]

line 29, Myringotomy and tube insertion combined with balloon eustachian tuboplasty has been demonstrated an effective and safe in the treatment of children with OME. Directly benefit from the ventilation tube, the curative effect was close during the period of tube retention. please discuss and cite Chen S, Zhao M, Zheng W, Wei R, Zhang B, Tong B, Qiu J. Myringotomy and tube insertion combined with balloon eustachian tuboplasty for the treatment of otitis media with effusion in children. Eur Arch Otorhinolaryngol. 2020 May;277(5):1281-1287. doi: 10.1007/s00405-020-05828-9. Epub 2020 Jan 30. PMID: 32002612.

Thank you. We added this sentence to the text of manuscript.

Moreover, even the enhancement of Eustachian tube function alone may contribute significantly to the curative process of OME. Balloon eustachian tympanoplasty combined with myringotomy and tube insertion was recognized as efficient intervention in the treatment of OME in children [17]

Methods

- apply the consort guidelines

- perform a flow diagram describing the protocol

Thank you. We tried to adjust consort guidelines for flow diagram of our retrospective, comparative, non-randomized clinical study (please see Supporting information – S1 Table)

- describe clearly selection criteria

Thank you. The inclusion criteria are given in the second paragraph of “Methods” section

- Age 3-17 years

- Follow-up period – 12 months

- Diagnosis of COM combined with adenoid hypertrophy

- The types of intervention – power-assisted adenoidectomy and type 1 tympanoplasty (at the same day or sequentially)

Results

- clear and well written

- always report both percentage and number or value

Thank you. We added numbers to Table 3

Discussion

- Adenotonsillectomy have been reported by several authors as a curative approach in children for both sleep apnea and behavioural disorders. Please discuss and cite doi: 10.1016/j.ijporl.2005.08.008. and doi:10.3390/children8100921.

Thank you. We added this data to the text of manuscript.

The existing data demonstrated the curative effect of adenotonsillectomy in pediatric patients with obstructive sleep apnea syndrome [19]. Furthermore, the performance of adenotonsillectomy improved the sleep-related quality of life and behavioral parameters in children with OSAS [20].

- Children with down syndrome demonstrated to undergo more repeat tube insertion (tti )if they were of younger age and if the indication for surgery up to 61%. please cite doi:10.1016/j.ijporl.2021.110811.

Nevertheless, in some unique patient groups several risk factors may contribute greatly to more repeat tube insertion (TTIs). For instance, the younger age and COME were recognized as the adjusted risk factors for TTIs in children with Down syndrome [22]

6. PLOS authors have the option to publish the peer review history of their article (what does this mean?). If published, this will include your full peer review and any attached files.

Do you want your identity to be public for this peer review? For information about this choice, including consent withdrawal, please see our Privacy Policy.

Reviewer #1: No

Reviewer #2: No

---

## [Editor Report · Decision Letter 1]

24 Feb 2022

Tympanoplasty and Adenoidectomy in Children: Comparison of Simultaneous and Sequential Approaches

PONE-D-21-32269R1

Dear Dr. Kassym,

We’re pleased to inform you that your manuscript has been judged scientifically suitable for publication and will be formally accepted for publication once it meets all outstanding technical requirements.

Kind regards,

Giannicola Iannella, M.D

Academic Editor

PLOS ONE

Additional Editor Comments (optional):

Very well written paper. Compliments
---

## [Editor Report · Acceptance letter]

2 Mar 2022

PONE-D-21-32269R1 

Tympanoplasty and Adenoidectomy in Children: Comparison of Simultaneous and Sequential Approaches 

Dear Dr. Kassym:

I'm pleased to inform you that your manuscript has been deemed suitable for publication in PLOS ONE. Congratulations! Your manuscript is now with our production department. 

Kind regards, 

on behalf of

Dr. Giannicola Iannella 

Academic Editor

PLOS ONE